# Challenges in Resource-Constrained IoT Devices: Energy and Communication as Critical Success Factors for Future IoT Deployment

**DOI:** 10.3390/s20226420

**Published:** 2020-11-10

**Authors:** Felisberto Pereira, Ricardo Correia, Pedro Pinho, Sérgio I. Lopes, Nuno Borges Carvalho

**Affiliations:** 1Instituto de Telecomunicações, 3810-193 Aveiro, Portugal; rjoao@ua.pt (R.C.); pedro.pinho@isel.pt (P.P.); sil@estg.ipvc.pt (S.I.L.); nbcarvalho@ua.pt (N.B.C.); 2Departamento de Eletrónica, Telecomunicações e Informática, Universidade de Aveiro, 3810-193 Aveiro, Portugal; 3Sinuta S.A., 3860-529 Estarreja, Portugal; 4ESTGV, Instituto Politécnico de Viseu, 3504-510 Viseu, Portugal; 5ISEL—Instituto Superior de Engenharia de Lisboa, Instituto Politécnico de Lisboa, 1959-007 Lisboa, Portugal; 6Instituto Politécnico de Viana do Castelo, Viana do Castelo, 4900-347 Viana do Castelo, Portugal

**Keywords:** Internet of Things (IoT), Wireless Sensor Networks (WSN), backscatter communications, Wireless Power Transfer (WPT), Energy Harvesting (EH), chipless devices, Simultaneous Wireless Information and Power Transfer (SWIPT), Wake-Up Radio (WUR)

## Abstract

Internet of Things (IoT) has been developing to become a free exchange of useful information between multiple real-world devices. Already spread all over the world in the most varied forms and applications, IoT devices need to overcome a series of challenges to respond to the new requirements and demands. The main focus of this manuscript is to establish good practices for the design of IoT devices (i.e., smart devices) with a focus on two main design challenges: power and connectivity. It groups IoT devices in passive, semi-passive, and active, giving details on multiple research topics. Backscatter communication, Wireless Power Transfer (WPT), Energy Harvesting (EH), chipless devices, Simultaneous Wireless Information and Power Transfer (SWIPT), and Wake-Up Radio (WUR) are some examples of the technologies that will be explored in this work.

## 1. Introduction

Words like smart or intelligent are nowadays widespread in technology-driven products. All around the world, but especially in the so-called developed world, there are people using smartphones, living in smart cities, driving smart cars, or being lit by intelligent lamps. The base behind all these creative innovations is Internet of Things (IoT), IoT, which based on non-scientific documents, was first introduced in 1982 when a modified coke machine was connected to the Internet to report how many drinks were inside and how cold they were [1]. Despite the broad set of applications and solutions, in the broadcast sense, IoT encompasses all devices and objects that can connect themselves, directly or indirectly, to the internet. The final idea is to allow the free exchange of useful information between multiple real-world devices. As Intel and HP convey in a more business-oriented language, IoT goal is to empower billions of existing devices with intelligence, connecting people to their content [2].

However, to achieve future predictions, several technological challenges need an answer, which may be divided into five significant issues: security and privacy, storage and cloud computing, energy, communication, and compatibility and standardization, illustrated in Figure 1.

Implementing security processes in IoT devices has a high level of complexity due to the limited capabilities available on the devices, their vulnerabilities are related to transport encryption, inadequate software protection, and insufficient authorization. These risks are threatening data privacy, which is a crucial aspect, especially when considering applications like health equipment and emergency services. The second issue regards storage and cloud computing that relates to the data generated by the increasing number of IoT devices. Currently, the used architectures for data centers are not prepared to deal with the quantity and diversity of data that billions of heterogeneous devices can generate. In addition, there is a need for new intelligent analytics mechanisms; just a small percentage of the data that is nowadays stored suffer a process of scrutinization to obtain valuable information. In the past, these mechanisms were mainly applied in a centralized architecture based on cloud structures; however, nowadays there is a clear tendency to prefer a certain degree of analytics in the edge devices—fog computing. These new decentralized architectures are most important to offload some of the computing and processing tasks occurring in the cloud, allow more efficient use of resources, provide highly responsive services, and enforce privacy policy [3,4]. Also linked to the number of IoT equipment is the problem regarding energy. Considered by many as one of the leading technological challenges for the next years, the way how energy is harvested, stored, and used will dictate the pace of evolution in the sector. Energy-efficient devices are imperative not only to enable more and more applications but also to make the existent ones greener and more environmentally friendly. Besides challenges associated with energy, it is also essential to consider the way how they communicate, the data rate of IoT terminals could vary from a few kbps to several Mbps, and distances that can vary, from a few centimeters to several kilometers. These specifications need to be taken into account when developing the communication system to achieve efficient use of power and spectrum, minimizing the interferences between communications. Lastly, the compatibility and standardization of IoT systems is a very desirable aspect once it would help all the other challenges already mentioned. Having certain norms would allow all actors to equally access and use, promoting the efficient use of IoT infrastructure and applications [5,6,7].

From the five challenges defined, energy and communication are the ones with a higher level of interest from an electrical engineering perspective. Addressing energy concerns can be done by exploring new sources for Energy Harvesting (EH), improving the efficiency of existing sources, putting together multiple sources, advancing in Wireless Power Transfer (WPT), developing charging and storage elements, or simply by ensuring the lowest energy consumption possible. By itself, each one of these elements can contribute to improve existing applications or to enable new ones. However, the final goal is to integrate all of them to significantly enhance IoT energy systems [8,9].

Regarding communication systems, there are multiple alternatives such as Wireless Local Area Network (WLAN), Bluetooth, Low Power Wide Area Network (LPWAN), as well as mobile networks systems more often used in commercial applications such as Wireless Highway Addressable Remote Transducer Protocol (WirelessHART) and International Society of Automation 100.11a (ISA100.11a), which are also used in industrial applications and use active radios [10,11]. In contrast, others like backscatter communication systems do not depend on the use of active radios. Backscatter communications are nowadays mainly used in Radio Frequency Identification (RFID) systems. However, the last years have brought many technological developments that can allow an application with this type of communication (wave reflection) on a broader range of devices and applications. Some areas of scientific interest are related to high order modulations, chipless devices, commercial protocols using backscatter (e.g., Bluetooth, LoRa), and digitally controlled modulations [12,13,14].

Grouping energy and communication characteristics allow the identification of three distinct IoT devices: passive, semi-passive, and active. This nomenclature, usually associated with RFID tags, describes the devices according to their need for batteries and radios. Equipment with no battery or radio are considered passive, the ones with battery but without radio are semi-passive, and the so-called active have both elements [15,16]. The presence or absence of these modules interferes with the device’s general characteristics, like communication range, processing capabilities, lifetime, or price. Figure 2 presents a comparison between the established types of IoT devices.

This work synthesizes useful information to provide a complete understanding of the energy and communication challenges for IoT. It scrutinizes some aspects of different techniques such as backscatter communication, WPT, Simultaneous Wireless Information and Power Transfer (SWIPT), chipless devices, and EH, that have been developed to overcome the mentioned IoT challenges.

The manuscript is organized as follows: Section 2 explores the technological developments that have been made in passive systems—backsctter communication, WPT, EH and chipless techniques; Section 3 points techniques used in semi-passive solutions—SWIPT and Wake-Up Radio (WUR); and Section 4 resumes some alternatives for active systems. Section 5 presents a discussion about the real application of passive, semi-passive, and active systems. The document ends with conclusions in Section 6.

## 2. Passive Systems

As previously defined, passive devices are characterized by the absence of battery and active radio. These remotely powered, fully passive transponders entirely rely on the power transmitted by the interrogator [17]. Due to these characteristics, these circuits are normally used simply as tag identifiers or, in some cases, as sensors, have a short range of operation and low processing capabilities. However, for a wide range of applications, its extended lifetime, placing flexibility, and low price are major advantages that justify the use of this technology. These types of IoT devices, generally associated with RFID tags, are based on technologies with a high maturity level and have been widely used in diverse applications fields, notably smart logistics, product traceability, and value chain optimization. In [18], the authors claim that RFID tags are “the next big thing” in management because these tags can optimize multiple business processes and, more importantly, the emergence of new intelligent or smart processes.

Backscatter communication, WPT, EH, and chipless techniques are the technologies that have been shaping passive solutions.

### 2.1. Backscatter Communication

Backscatter communication consists of reflecting the electromagnetic wave with a specific modulation, making it possible for the receiver to decode it. This principle relies on Modulated Scattering Technique (MST) to transmit unidirectional information from tag to reader. [19,20].

A passive system is usually composed of three different elements: an interrogator, a reader, and a tag. The interrogator is responsible for sending a Continuous Wave (CW) to the tag, which by its turn, modulates and reflects it. This reflection is received by the reader that demodulates the tag’s data. Backscatter communication can assume two different structures, monostatic and bistatic (collocated or dislocated). In monostatic architectures, the same device performs interrogator and reader functions. This structure allows simple implementation and cost reduction and is mostly used in commercial readers. The main drawback of this topology is that it results in higher path losses meaning shorter distances and less coverage (the round-trip, reader-tag-reader, the path loss is proportional to the 8th power of the distance). In bistatic, there are independent devices that allow separated locations. It has less path loss and allows the distribution of multiple interrogators or transmitter. It is possible to build a system with a centralized reader and multiple tags illuminated by multiple transmitters, which would allow a higher field coverage [18,21,22]. Figure 3 schematizes both topologies.

As mentioned, tags are responsible for modulating and reflect the incoming wave. The modulation process consists of changing the load impedance, and different load switching results in different reflection coefficients and thus distinct modulations.

Most commercial systems employ the simplest way of the Amplitude Shift Keying (ASK) scheme, the On-Off Keying (OOK). This 1-bit modulation consists in transmitting zeros and ones represented by the existence or absence of reflected wave. The reflection occurs when the circuit connects to a full-reflection load, i.e., open or short, and the absorption happens when it switches to a matching load. This binary scheme can further evolve to more complex ASK modulation based on different levels of reflected power [21]. Instead of using amplitude variations, it is possible to perform the modulation in different frequencies and phase reflections, i.e., Frequency Shift Keying (FSK) and Phase Shift Keying (PSK). Besides modulation that involves one particular component, it is also possible to perform backscatter modulation using multiple variations. The classic example is Quadrature Amplitude Modulation (QAM), which encompasses amplitude and phase modulations. Figure 4 illustrates some modulation schemes.

The modulation schemes represent the bases from which it is possible to define new scientific advances in backscatter communication. The interrogator replacement and the use of commercial readers are two areas of great interest once they can expand the use of backscatter to other systems.

The way to not use a dedicated interrogator is to take advantage of existing Radio Frequency (RF) signals e.g., radio, television, Wi-Fi, which is a technique known as ambient backscatter [23]. When compared with the traditional approach, the use of ambient backscatter has the clear advantage of not requiring a dedicated interrogator. However, it also presents some negative aspects like its signal stability and the limited modulation possibilities. As the interrogator has different purposes, it is not possible to control its stability, strength, or availability. It also needs to take into consideration the modulation (if any) of the source signal, i.e., using ASK modulation in a source signal with amplitude variation can make the backscatter communication unreadable [24,25].

The first work that introduced the concept of ambient backscatter used a TV tower signal broadcasting in the 536–542 MHZ range. The backscatter circuit was tested at different locations with the signal power varying from 24 dBm to −8 dBm, the maximum distance achieved between the tag and the TV tower was approximately 10.4
km [26]. Instead of using the RF signal from a TV tower, the work presented in [23] benefits from the Frequency Modulation (FM) radio to backscattering sensor information. The use of Wi-Fi as a source for ambient backscatter was explored in [27], achieving distances of 5 m between Wi-Fi source and modulation device, and transmission data rates of 60 kbps.

Besides ambient backscatter, the use of commercial readers can also create a new set of opportunities for backscatter solutions. Before going further on this topic, it is essential to note that the considered reader architecture is the one presented in the bistatic approach, by other words, the reader only needs to receive and demodulate information. The challenge goes to the tags that need to modulate information according to the reader’s protocol. In [28], Ensworth et al. present a Bluetooth Low Energy (BLE) backscatter prototype that performs band-pass FSK modulation on a CW carrier source. Readers such as Apple iPad Mini or Nordic Semiconductor nRF51822 could receive the signal at a distance of 13 m. The work presented in [29] also explores the BLE backscatter. Focused on allowing backscatter in LPWAN, Correia et al. [30], proposed chirp based backscatter modulator, which allows the reflection of LoRa preamble symbols with a Spreading Factor (SF) of 7 and 125 kHz of bandwidth. Also using LoRa, in [13], Talla presents a low-cost LoRa backscatter Integrated Circuit (IC) whit a power consumption of 9.25
μW. Figure 5 illustrates the schematics for ambient backscatter and proprietary reader.

Another important aspect of devices based on backscatter communication is their capabilities for use in sensing applications. Temperature, humidity, and other environmental parameters can influence dielectric properties and therefore change the natural response of a determined device. If characterized, this alteration can be understood as an indirect measurement [31]. By placing a RFID tag with two sets of parallel and series stubs on the side of a water tank, Capdevila et al. were able to create a water temperature sensor. In the same work, and using similar technologies, the authors also created a set of on–off tags to measure the water levels. To do that, the tags were placed inside a water tank. When submerged into water, due to the high permittivity, the tags did not respond to interrogation; when the water level dropped and the tags were exposed to air, they started responding, thereby creating the on–off sensor [20]. Taking advantage of indirect measurement, the work presented in [32] shows an adhesive sensor capable of monitoring sweat electrolytes. Besides the IC, the work uses a sensor that changes its impedance according to sweat levels, and the technology was created integrating both in the same device. Another example of combining backscatter techniques with sensing applications is provided in [33]. The authors used two RFID IC connected to two mercury switches with an antiparallel configuration. Using this architecture, Philipose et al. created a one-bit accelerometer: when the mercury is at one end of its container, one switch is closed and the other is open, which means that just one IC responds when interrogated; when the mercury is at the other end, the process is inverted creating the one-bit accelerometer.

### 2.2. Wireless Power Transfer

There are two types of WPT, near-field and far-field. In the near-field approach, power is transferred exploiting the magnetic field via inductive coupling between coils [34]. Far-field WPT uses electromagnetic waves with a wavelength between radio waves and infrared radiation to transmit energy at longer distances [35,36].

The main drawback of this method is the high power loss in the propagation stage. The first approach to overcome this loss is to increase the transmitted power; however, due to health impacts and regulatory issues, it is not desirable to increase the transmitted power and exceed the reference levels [37]. Given this, other techniques like waveform analysis and modeling of RF-DC converters, propagation modeling, and antennas specifically tailored for WPT schemes have been used in commercial devices, e.g., Ossia (Washington, DC, USA) [38] and Wi-Charge (Rehovot, Israel) [39].

Due to its possibility to power systems from relatively long distances, far-field WPT appears as an exciting approach when considering passive systems that do not have a battery. Using these techniques can also the deployment of passive Wireless Sensor Networks (WSN) creating new possibilities for IoT. The basic model for far-field WPT implies a transmitter and a receiver.

The more straightforward transmitter includes an oscillator, an amplifier, and an antenna. The oscillator generates the CW, the amplifier due to its gain increases the CW power, and the antenna radiates it [40]. Many studies have shown that the design of an optimum signal can increase the RF-DC conversion efficiency significantly and, consequently, the amount of power delivered to the load. This happens with Peak-to-Average Power Ratio (PAPR) that consists of creating a waveform with peaks of transmitted power without interfering with the average power transmitted. This technique allows the diode to be biased in situations that, with regular CW, it would not [40,41].

On the receiver side, there are always an antenna and a rectifier circuit (the set can also be called rectenna) but, it is also usual to have energy management and energy storage components. Generally, rectifiers are an association of Schottky diodes, output bypass capacitor, and a load resistor. Even though the design can assume multiple configurations, the single serial and shunt topologies are the largest employed [42]. Figure 6 presents different topologies for rectifiers. The charge pump rectifier is a single-stage voltage multiplier that ideally produces a voltage twice the peak voltage of the input signal. In the negative semi-cycles of the input signal, the second diode charges the second capacitor with the peak voltage of the input signal. When the input signal switches to its positive semi-cycle, the pre-charged second capacitor behaves like a voltage source in series with the input signal, thereby doubling the voltage of the signal that finally will be rectified by the first diode. This circuit produces more DC output voltage than single-diode rectifiers, but its efficiency is lower for low-input signals because there is an additional diode that influences its performance [43].

Figure 7 shows the RF-DC conversion efficiency of different rectifiers with different circuit topologies for frequencies between 450 MHz to 94 GHz, based on data published in [44,45,46]. Besides single-band converters, which are presented in most of the references depicted in the Figure 7, dual-band converters that harvest in two different sources are presented in [47,48,49,50], and a triple-band converter is presented in [51]. Some works presented a combination of different sources such as light, heat, electromagnetic waves, vibration, and others [52,53,54].

Most of the traditional rectifiers can only exhibit reasonable RF-DC conversion efficiency with a narrow input power range. The efficiency declines very quickly when the input power deviates from the operating range, limiting the wireless charging applications with considerable variations of input power. Thus, it is necessary to implement and design rectifiers with a wide operating input power range [55,56,57].

The work presented in [58] focuses on broadband matching networks and voltage-double circuits, it achieves a peak efficiency of 74.8% at 10 dBm at a center frequency of 1.3
GHz. On the other hand, Noughabaei et al. [59], explore dynamic and static bias compensation to decrease the transistors forward voltage drop, their main result is a sensitivity of −30.5 dBm at 915 MHz and maximum efficiency (which was not the primary goal) of 42.8% at −16 dBm. In [44], Valenta et al. present a table analyzing this discussion between efficiency and sensitivity.

Even though the techniques presented until this point can have a good impact in creating new WPT solutions, probably the big challenge in the topic is how to focus energy and improve the end-to-end transfer efficiency. To achieve this goal, transmitters need to be able to locate receivers in order to focus the transmitted energy.

One way to wireless locate devices is to measure the Received Signal Strength (RSS), the method consists in measuring the received signal strength and send it back to the reader so it can calibrate and understand the point with better results. However, this method can easily be corrupted by multipath effects, noise, and other environmental parameters that must be addressed by using complex strategies [60]. Another way is to measure the Time of Arrival (ToA), which consists of calculating the distance between the transmitter and the tag by computing the distance using Time-Difference-of-Arrival (TDoA) estimates. This method is commonly used in Global Positioning System (GPS), nonetheless, for common applications to have an accurate location the synchronization needs to be very precise [61]. Besides these methods, the Angle of Arrival (AoA) is also a widely used technique. It computes the target location by processing its received signal and understanding the AoA giving that information to the transmitter [62]. The main drawback of these techniques is that all of them require the receiver to have RF, and sometimes, computing capabilities which are not feasible for a large number of applications. The solution that has been proposed in this topic is the implementation of WUR [63]. As this technique can serve this and others purposes, it is further detailed in Section 3.

### 2.3. Energy Harvesting

Energy harvesting is known as the process by which energy is collected from different external sources to power low-energy electronics without any need for batteries. Solar power, thermal energy, wind energy, kinetic energy are some examples of external energy sources that may be considered [64]. There are uncountable energy sources that can be used, choosing the best energy source always depends on the final application. Due to their high availability and simple mechanisms, piezoelectric, solar, wind, electromagnetic, and thermal energy sources are the most used. However, there are also solutions like using microbial fuel cells [65] or chemical energy of glucose and oxygen [66], which may be useful. Besides the final application and the surrounding environment, it is also crucial to understand the power consumption profile. Only with all these variables, it is possible to create a solution that is resilient and autonomous from the energy point of view [67].

To have a closer look at some energy sources, the next lines detail four types of energy and the mechanisms used to extract DC power. Figure 8 illustrates the most used energy sources for energy harvesting.

Described as the sum of potential and kinetic energy, mechanical energy can be decomposed into two forms of energy: vibration and pressure. A usual way to harvest this kind of energy is by the use of piezoelectric materials.

To generate piezoelectric energy, it is necessary to have a transducer element attached to a flexible structure. The movement of the flexible structure will generate pressure or vibration in the piezoelectric element and consequently produce energy [68]. This movement can come from oscillations in buildings, bridges, human movement, or even vehicles and machinery. To demonstrate the capacity of piezoelectrics, Minazaa et al. [69], placed a piezoelectric transducer on the handle of a bicycle. The results show that the device is capable of feeding a sensor with a consumption of approximately 3.5
mW. In [70], Lallart performs a comparison between the energy density of different small-scale energy harvester, piezoelectric energy scored 35.4
mJc$m^{-3}$, higher than electromagnetic energy.

The main difficulty in harvesting energy from piezoelectric transducers is their high variability. When the oscillation is not constant, the piezoelectric elements can produce very high current peaks during short periods. Due to that, it is necessary to use a block of energy management that is responsible for rectifying and stabilize the energy delivered by the piezoelectric elements [71]. Usually, the block elements are a half or full bridge rectifier and capacitors.

Another widely available energy source is thermal energy. Converting heat to electricity is a process based on the thermoelectric effect, which explains that a junction of two dissimilar materials at different temperatures generates electrical energy. The devices that perform this conversion are known as Thermoelectrical Generators (TEGs), and have their base in three physical phenomena: the Seebeck effect is the voltage generated when temperature changes are maintained; the Thomson effect is the heating or cooling process which occurs when a current flows in the direction of a temperature gradient; the Joule effect is the heating of a conductor when an electrical current is present [72,73].

In its beginning, thermoelectric technology has been used mainly for home generating stations, but with technology developments, TEGs are already used to recover the heat generated by the exhausting pipes [72]. Now the focus regarding TEGs is miniaturization and make them available for low-power applications. Bahks et al. propose the use of TEGs as a solution for powering wearable devices by harvesting energy from the human body [74]. With an initial 35 mV of startup voltage, the work presented in [75] could generate 35 μW from the human body heat. Another example is to use TEGs as a wearable power supply for a pulse oximeter in a finger. The device can generate approximately 100 mW, which was enough to power the pulse oximeter that consumed 60 mW [76]. The schematic of a thermoelectric energy harvester used applied to the human body can be found in Figure 9.

Solar energy is a well-known source of renewable energy due to its high power demand. How to efficiently convert solar energy into usable electricity and how to store it are some of the hot research topics regarding this technology. However, and for the particularities of IoT, some research aims to bring solar energy to small scale, low power applications. An example of it is the work presented in [77], where the capture of ambient solar light powers a wireless sensor. Between solar energy and the wireless sensor, there is a DC-DC Buck that converts and regulates the magnitude of the voltage and a Maximum Power Point Tracking (MPPT) system. The last ensures the maximum power for any given environmental conditions due to its load optimization. Having a different scope, the work [78] presents an algorithm for solar energy prediction to provide an estimation of the upcoming energy profile. It uses this estimation to optimize system task scheduling overtime by allocating appropriate energy. The work considers the diurnal cycle and both seasonal and daily trends.

The biggest problems regarding solar energy are its instability and its almost non-existent energy indoors. Nevertheless, some works have been focusing on solutions that are capable of work in indoor environments. Even though they present different approaches, [79] and [80] apply different techniques to avoid the loss of energy during the conversion process. As the amount of solar energy in an indoor environment is significantly lower, these techniques are essential to allow its possible application.

Converting wind energy into electrical energy is something highly deployed when considering big turbines capable of generating megawatts. However, the generation of milliwatt electrical power from wind flow energy is relatively new [81]. This topic suffers from some ambiguity once it is difficult to define which dimensions should be considered suitable for the low-power applications. A work that reviews alternative power sources for remote sensors considers solutions with the blade’s size from 4 cm to 34 cm. These energy harvesters produce between 0.75
mW to 200 mW, and the number of blades can vary from 2 to 6 [67]. Showing the design and experimental validation of a wind harvester using a small turbine, the work presented in [82] is an excellent practical example. Its small turbine (5 cm of radius) is capable of supplying a microcontroller and a RF transceiver. The potential of using wind energy to supply low power systems is also demonstrated in [83], where an autonomous sensor is powered exclusively using energy generated from a wind turbine.

As wind energy, the technology of large-scale hydroelectric power generation is well established. However, it is not the focus of this work. Small-scale energy harvesters are not very usual, and the ones that exist are still big when compared with energy harvesters based on other sources of energy [67]. In spite of, the scarcity of works, it was possible to find some good examples. In [84], Taylor et al. designed a system capable of generating electricity from the flow of waters in the river and lakes. The integrated system, composed of an energy harvester and a storage unit, was tested in a flow tank. Instead of using rivers or lakes, the work presented in [85] uses irrigation pipes to generate energy, being able to produce 15 mA$h^{-1}$. The final goal was to provide energy to measure and transmit multiple environmental parameters in a vineyard.

Another option for EH is combining two or more sources of energy, frequently called hybrid energy harvesters. The main goal of these systems is to increase the amount of generated power, which can be especially attractive in situations with low or intermittent power [67]. As mentioned, one reason to create hybrid energy harvest systems is to overcome the discontinuities of power. Solar energy is an excellent example of this problem, and, due to that, the work conducted in [86] developed a system able to extract energy from photovoltaic devices and piezoelectric. Both harvesters have individual rectifiers, being after connected to a conventional battery. It provides energy to a hydrogen sensor, a sensor interface, as well as a microcontroller, and a transmitter. Instead of using sun and vibration, a different approach combines sun, wind, and water. Each source has its conditioning block and comparator. When the generated voltage is above a particular value, a comparator allows transferring the charge stored at that capacitor to the battery. If the voltage is below the required level, the same comparator turns off the converter to avoid battery discharge [85]. The work presented in [87] focuses on combining systems regardless of the kinds of energy. It consists of three sub-systems: energy harvesting, reservoir capacitor array, and control/charger. Each energy harvester charges its reservoir capacitor, and each capacitor smooths the dynamic range, which can cause the system to operate unreliably or fail. After that, the control and charge sub-system determines which power source, either the battery or the reservoir capacitor array, should power the load by comparing the terminal voltage and defined threshold voltage. This sub-system has a window comparator, threshold detector, load switch, battery charger, and battery protector. The same logic decides if the battery is charged by the capacitors or not.

### 2.4. Chipless Techniques

Traditional passive RFID tags never achieved a competitive cost to be the perfect substitute for the barcode. The target cost is less than 0.01 EURO, which could mean the best course is to eliminate their IC [88]. Besides its cost advantage, chipless tags can be fully printable, wholly passive, and environmentally friendly, which creates a new set of possible applications [89]. The big challenge regarding chipless RFID tags is how to perform data encoding without the presence of an IC. There are three main techniques to approach this problem: Time Domain Reflectometry (TDR), spectral signature, and amplitude/phase backscatter modulation.

One way to implement TDR is through the use of Surface Acoustic Wave (SAW). SAW devices follow the same principle of piezoelectrics, converting mechanical stress into electrical polarization and vice versa. The process starts with the tag’s antenna receiving the electromagnetic wave emitted by the reader, which is converted into a SAW using an interdigital transducer. The SAW propagates along the tag’s substrate, generating a unique time-varying acoustic-wave pulse chain when it encounters the reflections grating. The pulse is once again converted into an electromagnetic wave by the same interdigital transducer. Since the SAW is 100,000 times slower than the original interrogated pulse, all environmental echoes are sufficiently dissipated, making it possible to analyze data without interference. Information encoding happens due to the location of the reflections; different locations create different echos and, consequently, different data [88,90]. In Figure 10 it is possible to see the illustration of this principle.

The main application for SAW devices has been identification proposes; however, if they can either alter the velocity of the acoustic wave or deform their mechanical structure, they can also work as sensors. One example is the measurement of temperature using SAW techniques. In [91], Kang et al. achieved an accuracy of ±0.3 °C in the temperature range 0–40 °C. Another example using SAW is in strain measurement, which can use tiny sensors (less than 7 mm). The sensor has different resonant frequencies to different strains: the higher the resonant frequency lower the strain [92].

Besides SAW it is also possible to develop chipless TDR using RF designs. The work developed by Mandel et al. in [93] shows two different designs: one based in line modulators and the other on power dividers. The line modulators are implemented using delay line sections interrupted by modulation sections, where these modulation sections are re-configurable to obtain different IDs with the same tag. In the design based on power dividers, the interrogation pulse approaches an asymmetric shape and is split into two different fractions. Each fraction has different reflections causing a unique reflection. Like the line-based modulator, this design is continuously interrupted to maintain its reconfigurability. In addition, consisting solely of micro-strip structures, the work presented in [94] shows a 16-bit modulator. As the authors explain, a specific reader is needed to achieve a high modulation scheme (considering TDR techniques). To take advantage of the 16-bit modulator, the work presents a reader in which the interrogation signal is a frequency-modulated CW generated by a Voltage Controlled Oscillator (VCO) with a center frequency of 7.825
GHz and a bandwidth of 1.25
GHz.

Spectral signature relies on materials with variable electrical conductivity, dielectric permittivity, or permeability, to create different modulations. The modification in their physical properties would present changes in their amplitude, frequency, or phase, encoding the information in the resonant structures. These tags are fully printable, robust, and also have higher data storage when compared with other techniques, but they require a broad spectrum to operate. One way to spectral sign a tag is by using a slot structure to encode information. This structure has slots with different lengths on the same tag to generate different resonant frequencies. The absence of a particular slot, and consequently, the absence of resonant frequency, can also be understood as coding. Besides being used for identification, this kind of tag can also serve monitoring purposes [95].

In [96], the authors propose a fully printable chipless RFID tag operating in the frequency span 2 GHz to 4 GHz. The tag is made of 20 scatterers giving it 20 bit of coding capacity for a dimension of 70 × 25 mm${}^{2}$. This has achieved controlling the physical dimensions of each scatterer to create a sharp peak at specific frequencies. Each resonance frequency is understood as a one by the reader and its absence as a zero. Using a different design, but based on the same principle, in [97] Song et al. presented an 8-bit chipless device operating in the frequency band from 5.00
GHz to 5.70
GHz. Deng et al. presented a chipless tag for low-cost and long-term humidity monitoring. The tag was composed of six slots: the first three with different lengths working as an encoding unit to store ID information, and the next three with the same length, and covered by a sensitive material. As the dielectric constant and conductivity of the material vary according to the ambient humidity, it can work as a sensor [98]. Instead of using multiple slots and its absence for encoding information, it is also possible to encode the data in one unique resonator. To do that, it is necessary to apply frequency shifting techniques, where each resonance would represent a different ID. This method was applied in [99], where authors designed different paper tags to evaluate the possibility of measuring humidity. Results show that for a variation of 20–90% in humidity, the resonator frequency changes from 215 MHz to 195 MHz.

As the previous examples indicate, attributing sensing capabilities to spectral signed tags is another research topic of great interest. These sensing capabilities are typically associated with smart/sensing materials, which are materials that change transmission responses of microwave devices under the influence of varying physical parameters. From a first perspective, and due to its extensive use in circuits, conductive materials would be the right materials to be “smart.” However, these materials have low sensitivity to environmental changes, which makes them difficult to use for sensing proposes. On the other hand, semi-conductive materials have noticeable dielectric and conductive property changes with specific physical parameters, characteristics required for RF sensing applications. Inside smart materials, it is also possible to classify the materials according to the desired measurement [100]. Figure 11 presents that relation for the main smart material; however, there are others like graphene or nanowires.

The last technique to implement chipless tags approached in this work uses amplitude/phase backscatter modulation. This method encodes its information by varying the amplitude or phase of the backscatter signal, which requires less bandwidth and has a simple architecture when compared with previous methods. The use of lumped/chipped components, which increase the tag’s cost, can be considered its main drawback [101]. The use of amplitude backscatter modulation in chipless tags consists of different power reflection to encode different information. In [102], the authors used this technique over silver nanoparticles-based ink to create an ASK modulation. The final results show that more than 6 bits can be encoded. Like the previous methods, the amplitude modulation can also perform sensing capabilities.

Gonçalves et al. [14], explored the cork’s ability to absorb water to create a chipless sensor that varies the reflected power according to the cork humidity. The power level between a wet and a dry condition varies more than and it can be read at a distance of 3 m. In addition to humidity, it is also possible to measure other parameters like temperature. A possible approach is to use a transducer based on a capacitor with a micro-fluid channel that changes its permittivity according to the temperature. In this way, temperature changes would change the reflected power. The work presented in [103], uses this technique for a frequency of 29.75
GHz resulting in a variation of between 24 ${}^{\circ }C$ and 33 ${}^{\circ }C$. Instead of encoding information in the reflected power, phase modulation encodes the reflecting signals at different phases, see Figure 12. To do that, the transponder needs to have different load terminations and high impedance stubs. Balbin et al. [104], tested this theory developing three different patch antennas and measure their response with and without a stub length of 10.9
mm. The results have shown that it is possible to change the reflection by a 30 ${}^{\circ }$ phase shifter, changing the unique code. In [93], Mandel et al. presents the development of a high order PSK chipless tag. The tag is composed of line sections which can be removed and added (glued) to create a re-configurable tag that can realize different ID. The final results show a chipless device capable of 16-PSK.

## 3. Semi-Passive Systems

Semi-passive systems are characterized by having tags powered exclusively or partly by batteries and the absence of active radios, relying on a passive reflection modulation to communicate with the interrogator [17]. When compared with passive tags, semi-passive have a much higher computational potential, however at the cost of having a battery that eventually needs to be recharged and replaced. These characteristics are especially interesting in quality-oriented tracking and tracing [106]. Similary to passive solutions, semi-passive uses backscatter techniques to communicate, and can use WPT or EH to charge its batteries. Besides these technologies already approached in Section 2, semi-passive tags can have methods to receive information and specially designed WUR solutions to use their energy in the most efficient way possible.

Regarding possible applications, semi-passive tags can be applied to monitoring food conditions during the whole supply chain. It is of particular interest in food that needs to be in controlled environments and is very sensitive to temperature or humidity variations. Besides that, these devices can also be beneficial in medicines or other medical products that need strict monitoring, or to any product that needs continuous monitoring and communication at specific locations [106].

### 3.1. Simultaneous Wireless Information and Power Transfer

As the name illustrates, SWIPT is a technique used to transmit information and power in the same signal. This method is mainly used in semi-passive systems due to their processing capabilities to demodulate the information and their need to recharge batteries. Even though it is also possible to implement it in a passive solution. Regarding its potential applications, SWIPT is pointed to be very interesting to apply in structure and healthcare monitoring, as well as building automation [107].

SWIPT systems are generally formed by a Hybrid Access Point (HAP) and multiple nodes ready for EH and receive information [108]. To implement these solutions, different receiver architectures have been proposed, including an independent receiver, time switching, power splitting, and antenna switching.

The independent receiver architecture implementation uses simple components and techniques once each element works autonomously. It also allows for EH and information decoding concurrently. The main drawback of using two independent circuits is due to its non-efficient use of resources; in the decoding channel, it despises all the energy that arrives and vice-versa [109]. Figure 13 illustrates the independent receiver architecture. This architecture is used in [110], characterizing the tradeoff between the maximal energy transfer and information rate.

Also known as the co-located receiver, the time switching architecture share the same antenna for EH and information decoding. The receiver uses a switch that, according to precise instructions, selects the EH or information decoding path. This kind of system requires scheduling and time synchronization to ensure the choice of the right circuit. Figure 14 presents the time switching architecture. This approach is followed in [111], where the authors aim to optimize the energy efficiency whilst satisfying the constraints on maximum transmit power budget, minimum data rate, and minimum harvested energy per-terminal. The work focuses its application in 5G communications.

Another way to design a SWIPT receiver is by using the power splitting architecture. This method consists of dividing the receiver signal into two power streams, making it possible to simultaneously decode information and harvest energy. The power level on each channel should ensure the maximum energy available in the EH circuit without compromising the decoding. This architecture, represented in Figure 15, is recognized by achieving the best trade-off between information rate and amount of RF energy transferred [109]. In, [112], Camana proposes a system where a base station sends information and transfers radiofrequency energy to a receiver equipped with a power-splitting structure. One of the focus is to optimize the base station transmitted power and information always guaranteeing the receiver side.

Simpler than time switching or power splitting, antenna switching is another possibility to enable SWIPT. It can work with multiple antennas, where each antenna can be re-configurable between the two modules to achieve the best overall performance. For example, antenna switching architecture can use a set of antennas dedicated to EH and other set focused on information decoding [109]. Figure 16 clarifies the antenna switching architecture. The work presented in [113] aims to show that the antenna switching in the transmitter and receiver side leads to a win-win situation.

### 3.2. Wake-Up Radio

The most basic approach to reduce the energy consumption of any device is to apply a duty cycling algorithm. It consists of switching from active mode to sleep mode, minimizing the equipment power consumption. However, as almost all devices need to receive and transmit information, it is crucial to ensure that when some device is transmitting, the receiver is awake and ready to listen. There are three primary schemes to achieve that: pure synchronous, pseudo-asynchronous, and pure asynchronous [114].

In pure synchronous, the devices are pre-synchronized and are recurrently re-synchronized. This synchronization takes time and consumes considerable energy. Pseudo-asynchronous consist of one device waking and transmitting a long enough preamble to coincide with the wake-up schedule of the destination device. If for some reason, the destination does not respond, this preamble can have considerable energy consumption. By last, pure asynchronous rely on WUR techniques, the devices are in sleep mode and can be woken by the transmitter using a very-low power wake-up receiver [114].

Depending on their characteristics, WUR can be classified into passive and active.

In the same way, passive systems do not have any power source, passive WUR also does not. As they rely only on a RF rectifier, they can not distinguish between wake-up signals or a signal from other RF activity. They are also unable to receive any commands. Besides that, these circuits are also very limited in their range; their main use is short-range applications [115]. The circuit for a passive WUR consists of an antenna, matching network, rectifier, comparator, and the associated reference generator. The rectifier architecture is the same used for schemes of EH or WPT. However, instead of delivering the DC power to a storage element, the output voltage is used to power the reference generator and a low-power comparator. A fraction of this DC output is then compared with the generated reference and, if it is the case, the device is waked-up. Kamalinejad et al. follow this approach in [116], where simulations show a sensitivity of 33 dBm for zero power consumption. The authors of [115] implemented in practice this method. They used commercial RFID readers as transmitters and achieved 100% of wake-up probability for 4 m, dropping down to 0% beyond 5 m. Figure 17 shows the presented architecture.

The second group of RF wake-up receivers is known as active once they have internal power sources to power partly or fully supply their components. The ones in which the power source supplies only a minimal part of the components are also called semi-passive WUR and are the most common in literature. Generally, in this type of circuits, the rectifier and the envelope detector are just powered by the incoming RF signals while the energy storage unit powers the comparator. Figure 18 serves as an example of a base active architecture. Different from passive architecture, these methods can decode commands and information, allowing them to avoid unnecessary wake-up when other RF signals are present. In [117], the authors built a WUR for the frequency of 2.4
GHz. They implemented a zero-bias Schottky voltage doubler as an envelope detector and a three stages decoder based in an amplifier, Pulse Width Modulation (PWM) demodulator, and a comparator. The work achieved a power consumption below 20 μW for an incoming signal of approximately −53 dBm. More focused on achieving the lowest power consumption possible, Boaventura et al. [118] achieved near 10 μW of power consumption for a sensitivity of −35 dBm. The work used a two voltage multiplier and a micro-power comparator.

## 4. Active Systems

Defined by having an active radio, active systems architectures have high computational power and multiple ways of communication. The main disadvantage of these devices is its need for a constant source of power, which many times is a battery that may or may not be rechargeable. Different from the passive and semi-passive systems, that use backscatter techniques to communicate, active systems can take advantage of their active radio. This allows excellent adaptability for different applications, which should be specified in terms of the required data rate and range of communication. It is also possible to consider a third variable that is energy consumption; however, it is a result of the previous. Higher data rates and longer ranges result in higher energy consumption. According to these parameters, it is possible to group active systems in three categories, short-range, mobile networks, and long-range [119,120]. Figure 19 presents a schematic of this organization.

### 4.1. From Short- to Long-Range Communications

Based on RFID characteristics, Near Field Communication (NFC) is a wireless communication interface that works over small distances, typically shorter than 10 cm. Devices using NFC can create their own RF field or retrieve the power generated by another device. The former happens when the device is sending data, while the latter happens when it is receiving. The communication is performed at 13.56
MHz with ASK modulation, with an approximate power consumption of 60 mA. This technology is widely implemented all over the world in smartphones and other tracking devices. The set of applications ranges from simple identification to micropayments [18,121].

BLE is an evolution of the traditional Bluetooth technology more adapted to IoT solutions. In its latest version, BLE can perform communications at a maximum distance of 1000 m in an open field, and near 400 m indoors. It works at the 2.4
GHz Industrial Scientific and Medical (ISM) band, and it can be used to send data up to 2 Mbit$s^{-1}$. Its power consumption depends on the chip used; however, it usually is between 6 mA and 10 mA. In terms of applications, BLE is widely used in devices that need to communicate with smartphones [122,123].

WirelessHART is a communication protocol built upon IEEE 802.15.4 developed by the Hart Communications Foundation. It uses the 2.4
GHz band to create a mesh network where each participant station serves simultaneously as a signal source and a repeater. This process automatically adapts in the presence of interference from other networks, self-interference, or radio shadows, and it creates a reliable mechanism that occurs until the message reaches the final receiver. The communication is coordinated with Time Division Multiple Access (TDMA), synchronizing the network participants in 10 ms time-frames. WirelessHART modules can have a transmit current of 57 mA when transmitting at 14 dBm, and a current of 28 mA when receiving in high-gain mode. The overall link budget can achieve 122 dB and the data rate 250 kbit$s^{-1}$ (IEEE 802.15.4). Regarding its distance of communication, it is pointed to be 300 m indoors and up to 2000 m in line of sight. As mentioned, this technology is particularly interesting due to its resilience and delivery capabilities, which are crucial aspects in Industrial Internet Of Things (IIoT) [124,125,126]. Another protocol widely used in IIoT is ISA100.11a, which is by far the most complex and customizable standard. The network works based on contracts between devices that need to be signed before a transmission can occur. These contracts can be periodic or non-periodic [10]. As they are based on very similar hardware, ISA100.11a RF modules have identical metrics in power consumption, range, and link budget [127]. The big difference between protocols resides in its goal, while WirelessHART is designed to establish and assure interoperability between manufactures, ISA100.11a has many options for users to ensure compatibility between devices in the network [10]. Nevertheless, there are also dual-standard solutions that increase flexibility, management capacity, and reduced power consumption due to a decrease in network complexity [125].

The following short-range interface under analysis is IEEE 802.15.4 (standard which defines the physical and medium access control layers of the well-known ZigBee), which is a communication protocol created to communicate small amounts of information. It uses an open standard protocol that can perform communications at a maximum distance of 100 m with a maximum throughput of 250 kbit$s^{-1}$. Its power consumption can vary between 10 mA and 12 mA, and it also operates in the 2.4
GHz ISM frequency band. Zigbee is mainly used in home automation as well as in monitoring and control in industrial processes [85,128].

The most widely applied short-range communication interface is WLAN 802.11, designed to maximize data rates. In its 802.11ac version, it can theoretically reach 1.3 Gbit$s^{-1}$ and is named Gigabit Wi-Fi. It can operate at 2.4 and 5 GHz, and even though it is mainly used indoors, it can reach up to 100 m when in free space. The power consumption of WLAN 802.11 modules highly depends on the power they are transmitting and the rate they are achieving, reaching values of 240 mA [129,130].

Mobile networks are RF networks distributed over land areas called cells, where each cell has at least one corresponding base station. The system uses different frequencies between neighbor cells to avoid interference. The key feature of mobile network services over all the others is that users can experience ubiquitous and continuous coverage [131].

The first generation of wireless telephone technology was based on circuit-switched technology and designed only for voice. Digital signals in the 2G later replaced the analog radio signals used in 1G. Besides signal digitization, 2G also allowed Short Message Service (SMS) and was the origin of Global System for Mobile Communications (GSM). The transition between 2G and 3G was done gradually with some considerable improvements over the 2G original network. In the first 3G network, the main goal was to provide enhanced clarity and perfection as in real conversations; however, in its more recent releases, it could provide broadband access [131,132]. The fourth-generation of mobile networks was designed to offer broadband internet experiences relying on all-Internet Protocol (IP) communication instead of circuit-switched [133].

Even though the existing and widely applied mobile networks have some support for IoT-based services, they lack full optimization for it. Until this point, instead of having a mobile network designed for its requirements, IoT services have adapted to existing conditions. Examples of it are IoT devices that use SMS to send small amounts of data or voice beeps to send alerts and commands. 5G is expected to significantly change this scenario by connecting billions of smart devices, allowing a massive IoT with devices mutually interacting and sharing data without any human assistance. Even though its technical details are not entirely defined, 5G for IoT can bring high data rates, very low latency, highly scalable and fine-grained networks, long battery lifetime, high connection density, and mobility.

Sigfox is a proprietary ultra-narrowband technology designed to transmit small amounts of data over time. It allows a maximum of 140 messages per day, a payload of 12 bytes, and transmission throughput of 100 or 600 bit$s^{-1}$ depending on the region. This low data rate allows high power savings, ensuring IoT devices that use this interface have efficient use of their batteries and a distance of communication that can reach 10 km. When transmitting, each Sigfox module can consume up to 49 mA. At the moment of writing, Sigfox does not have worldwide coverage, but it is highly implemented in European and American continents [134,135].

In the same line as Sigfox, LoRa is also an interface designed to ensure the maximum distance of communication possible. This is possible due to the Chirp Spread Spectrum (CSS) techniques, which allow LoRa chips to have a sensitivity of −146 dBm, decoding information below the noise threshold. Its data rates can vary from 250 bit$s^{-1}$ to 50 kbit$s^{-1}$, and power consumption can reach 39 mA. It operates at unlicensed ISM bands: 868 MHz in Europe, 915 MHz in North America, and 433 MHz in Asia. The maximum distance of communication between LoRa devices, 766 km, was set during a test, including atmosphere balloons. However for general IoT the expected range cannot go much further than 10 or 15 km [119,136]. Narrow Band (NB)-IoT is a technology projected to coexist with GSM and Long Term Evolution (LTE) under licensed frequency bands, 700, 800, and 900 MHz. It occupies a frequency band width of 200 kHz, which corresponds to one resource block in GSM and LTE transmission. Regarding its data rate, it is limited to 200 kbit$s^{-1}$ for downlink and 20 kbit$s^{-1}$ for uplink, and each message can carry 1600 bytes. As it operates in the licensed spectrum, the network is not subject to unwanted interference. Different from LoRa or Sigfox, NB IoT also handles low bidirectional latency, at the expense of increased energy consumption. When transmitting, its power consumption can vary from 70 to 220 mA according to the transmitted power [119,137].

Table 1 presents an overview of the differences and unique characteristics of some active communication systems. From this table, it is possible to understand which systems are more suitable for long-range, high data rates, or low power communications.

### 4.2. Active Radios Powered by Coin Cell Batteries

Due to their power consumption and characteristics, BLE, LoRa, and SigFox have been used in multiple IoT applications. BLE has a lower power consumption and higher data rates but presents a shorter distance of communication when compared with LoRa and SigFox. This difference makes BLE more suitable for applications that require high levels of data transmission or a higher communication frequency. LoRa and SigFox are preferable for applications requiring long distances. As both technologies have very similar energy profiles, only LoRa is studied in this context.

Besides this distinction, the following lines explore the possibility of developing a real application scenario using BLE or LoRa communications powered by a coin cell battery. The application would be on an agriculture farm that requires environmental monitoring (e.g., atmospheric temperature and relative humidity) each 5 min, corresponding to 20 messages per hour. For higher-range applications, the BLE packet structure takes 376 μs before transmitting its data unit. As it contains the payload, the data unit size depends on the amount of data being transmitted, which for the required application can be considered to be 16 bytes (8 bytes for atmospheric temperature and 8 for relative humidity) [138]. In this condition, the transmitting message would take approximately 1.7
ms as detailed in Figure 20.

By its side, based on [139], a LoRa packet using a SF = 7 and a payload of 16 bytes would take approximately 40 ms. Increasing the SF to achieve the maximum distance can increase the packet duration up to 1.2
s. For further calculations SF = 7 is considered. Figure 21 illustrates the LoRa package.

To compare the technologies, technical information from Nordic Semiconductor nRF52810 [123] and Microchip RN2483 [136] was collected from datasheets. Both devices were considered to be equipped with the same sensor (SHTC3 from Sensirion), consuming 0.5
μA [140], and supplied by the same battery Panasonic BR2032 which has 200 mAh for a nominal voltage of 3
V.

The BLE nRF52810 chip consumes 7.5
mA when transmitting and 0.6
μA when in sleep mode. On the other hand, the LoRa RN2483 chip uses 39 mA when transmitting and 1.6
μA when in sleep mode. Using the coin cell battery and the predefined energy profile (transmitting 20 messages per hour) it is possible to estimate how long each solution could operate in a real scenario.

Using the total packet time information, Equation (Equation 1) calculates the time each technology spends in active and sleep mode.
(1)$$\begin{array}{c} BLE_{Tx}=1.7\;ms\times 20\;\mathrm{messages}=34\;ms\;h^{-1}=0.034\;s\;h^{-1}\\ BLE_{Sleep}=3600\;s-0.034\;s\approx 3600\;sh^{-1}\\ LoRa_{Tx}=40\;ms\times 20\;\mathrm{messages}=800\;ms\;h^{-1}=0.80\;s\;h^{-1}\\ LoRa_{Sleep}=3600\;s-0.8\;s\approx 3599.2\;s\;h^{-1} \end{array}$$

Using the values calculated in Equation (Equation 1) it is possible to calculate the average consumption of each device. The results are presented in Equation (Equation 2)
(2)$$\begin{array}{c} BLE_{ChargeConsumed}=BLE_{Tx}\times 7.5\;mA+BLE_{Sleep}\times 0.0006\;mA+0.0005\;mA=0.0012\;mAh^{-1}\\ LoRa_{ChargeConsumed}=LoRa_{Tx}\times 39\;mA+LoRa_{Sleep}\times 0.0016\;mA+0.0005\;mA=0.0108\;mAh^{-1} \end{array}$$

The following Equation (Equation 3) calculates how much time can each technology operate powered by a single coin battery with a electric charge of 200 mAh.
(3)$$\begin{array}{c} BLE=\frac{200\;mAh}{BLE_{ChargeConsumed}\times 24\;h\times 365\;\mathrm{days}}\approx 20\;\mathrm{years}\\ LoRa=\frac{200\;mAh}{LoRa_{ChargeConsumed}\times 24\;h\times 365\;\mathrm{days}}\approx 2\;\mathrm{years} \end{array}$$

At this point it is important to consider these calculations were based on simplistic theoretical information, only considering the energy consumed during the transmission and sleep periods. The transition between states, temperature influence, and others would have a considerable impacts on both results (a battery self-discharge rate of 3% would reduce the lifetime of the BLE device over 5 years). However, as this study intended to show, it is possible to build an IoT application based on BLE or LoRa technology powered just by a coin battery.

## 5. Discussion

Until this point, this work synthesized useful information to provide a complete understanding of the energy and communication challenges for IoT. The solutions and technologies have been divided into passive, semi-passive, and active to provide adequate organization. However, in the real world, this distinction is not so clear, and each case should be evaluated as a particular situation.

Applications where communications occur at short distances can be handled by passive solutions, combining backscatter communication with WPT or EH. However, there are certain applications that, although require short-distance communications, have high power demands. In these cases, and if WPT or individual EH is sufficient, a hybrid EH solution may be considered.

When possible, the use of passive solutions is always recommended due to its battery-less structure. However, if it is not an option due to high power demand, semi-passive systems can be considered. Using a battery allows independence from the receiving or harvesting system, and it ensures stable and uninterrupted flow of energy. To enhance its energy efficiency, the final solution must be designed to operate in active modes for the shortest time possible. One interesting method to ensure maximum energy efficiency is to implement WUR solutions. These techniques have architectures as simple as an RF-DC.

To avoid battery replacement, recharging, and also to ensure a bi-directional communication system, SWIPT techniques can also be implemented. It is important to note that SWIPT requires demodulating information on the tag, which is possible in semi-passive solutions due to its stable energy flow and high computing capabilities. If possible, the use of amplitude modulation, more concretely OOK, can simplify the whole process. The demodulation can be built with a peak detector connected to a digital or analog pin.

Due to the inconvenience of having a specific reader, some solutions for short-range communication rely on active devices. The most straightforward technology is BLE, which has sleep and active modes with considerably low power consumption, making it possible to be powered by a small coin battery. The lifetime of these solutions highly depends on the frequency of communications. In applications that only require a few communication per hour, coin batteries can last for some years.

On the other hand, some applications need to communicate at long distances. Up to this point, LoRa and SigFox have proven to be very interesting technologies to do that; both have ranges of a few kilometers with relatively low power consumption. Like BLE solutions, an optimized tag based in LoRa or SigFox can operate with a coin battery.

## 6. Conclusions

This discussion presents a review of techniques that have been developed to overcome the IoT challenges regarding energy and communication. It encompasses passive, semi-passive, and active technologies.

Passive solutions have been focusing on ways to improve their communication, harvest energy, and lower their cost. Regarding communication techniques, the goals are the increase the communications range, by exploring different architectures and improved waveform designs, achieve high data rates, by developing different modulations techniques, and reuse existing RF signals and proprietary readers. The energy concerns are related to the increase of the WPT efficiency and develop EH solutions capable of power devices without interference or failures. The cost challenge has been approached by developing new chipless tags.

Semi-passive systems have taken advantage of the advances in passive solutions. Besides that, some works have also have been focusing on ways to improve how semi-passive devices use energy and receive information. In SWIPT, different architectures have been proposed allowing a better fit between application requirements and used techniques. WUR has also been evolving, by providing a trade-off between robustness and energy efficiency.

Regarding active systems, recent advances have allowed different commercial solutions that are already enabling a wide range of options that can be selected according to the specifications of each application. In this case, the trade-off is between range, data rate, and energy consumption.

Having the knowledge and the understanding of these three different types of solutions provides a set of tools that should be used to create the most suitable solution for each application.

## Figures and Tables

**Figure 1 sensors-20-06420-f001:**
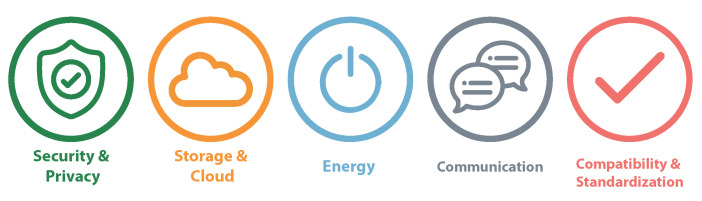
IoT technological challenges.

**Figure 2 sensors-20-06420-f002:**
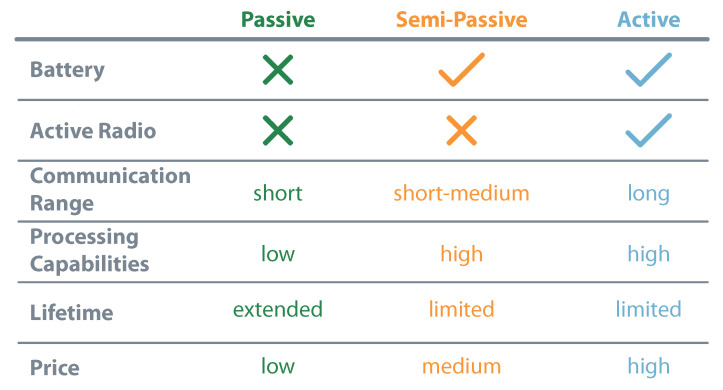
IoT characterization considering energy and communication.

**Figure 3 sensors-20-06420-f003:**
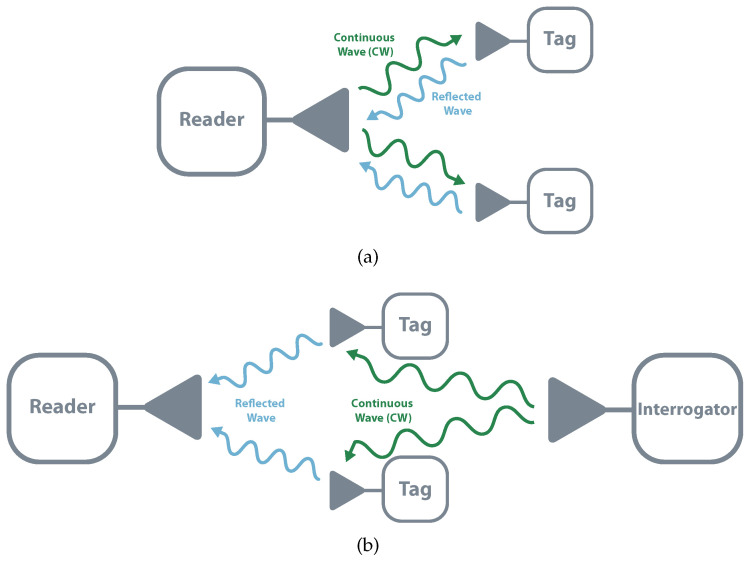
Backscatter architectures. (**a**) Monostatic (**b**) Bistatic.

**Figure 4 sensors-20-06420-f004:**
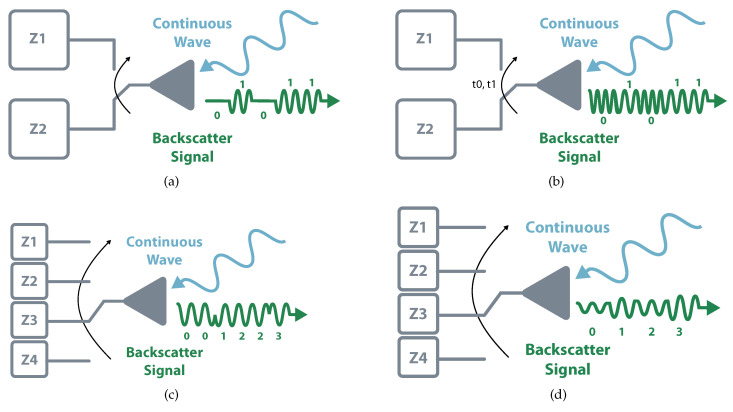
Modulation Schemes. (**a**) On-Off Keying. (**b**) Frequency Shift Keying. (**c**) Phase Shift Keying. (**d**) Quadrature Amplitude Modulation.

**Figure 5 sensors-20-06420-f005:**
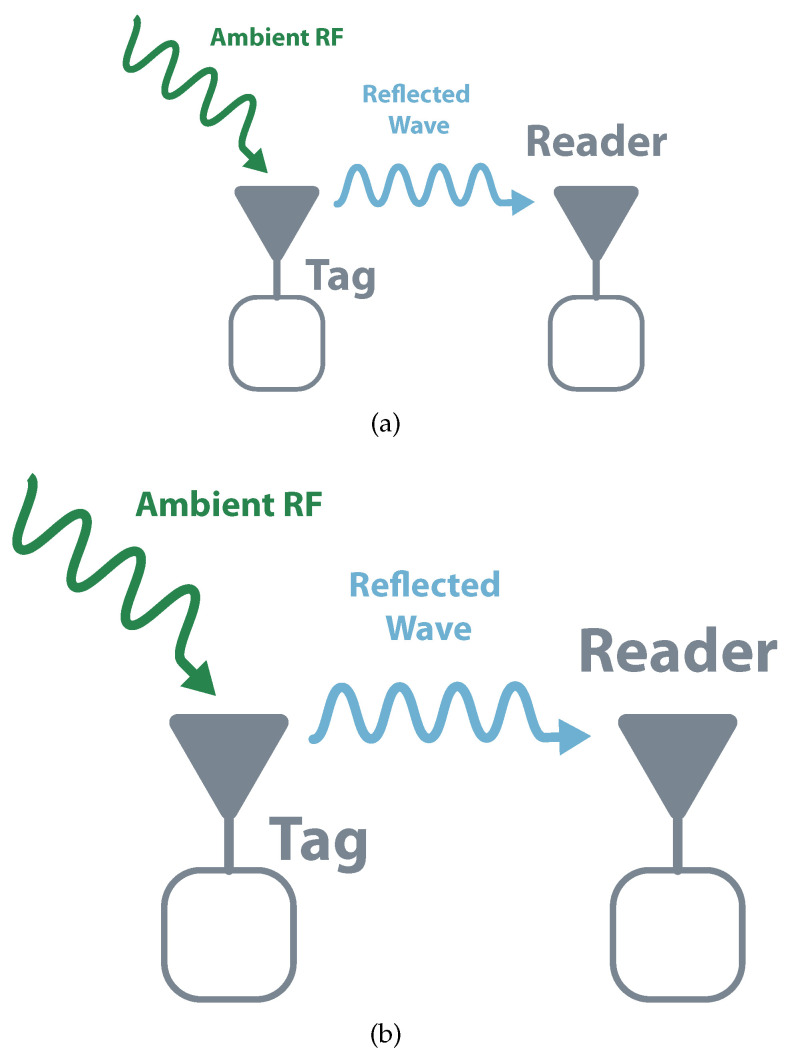
Backscatter systems without dedicated interrogators or readers. (**a**) Ambient backscatter (**b**) Proprietary reader.

**Figure 6 sensors-20-06420-f006:**
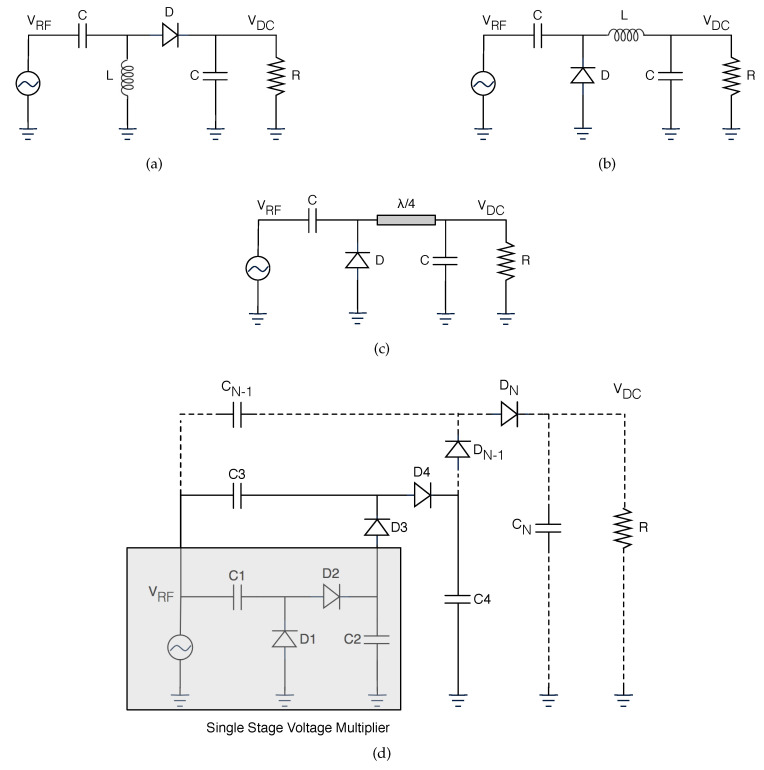
Different rectifier topologies. (**a**) Series diode rectifier. (**b**) Shunt diode rectifier. (**c**) Shunt diode with λ/4 stub. (**d**) N-stage Dickson voltage multiplier.

**Figure 7 sensors-20-06420-f007:**
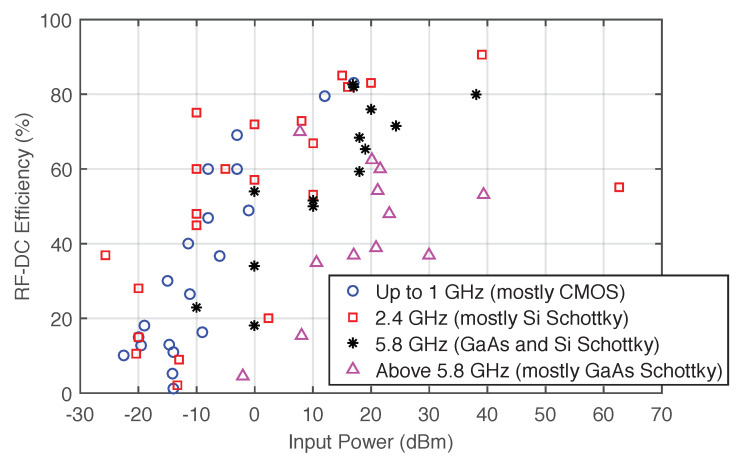
State of the art in RF-DC converters, with different topologies, from 450 MHz to 94 GHz. Adapted from [44].

**Figure 8 sensors-20-06420-f008:**
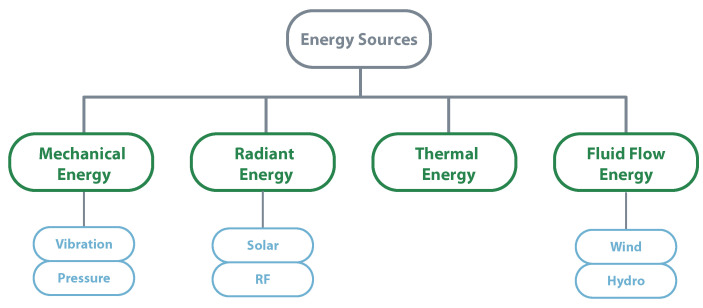
Most used energy sources for energy harvesting.

**Figure 9 sensors-20-06420-f009:**
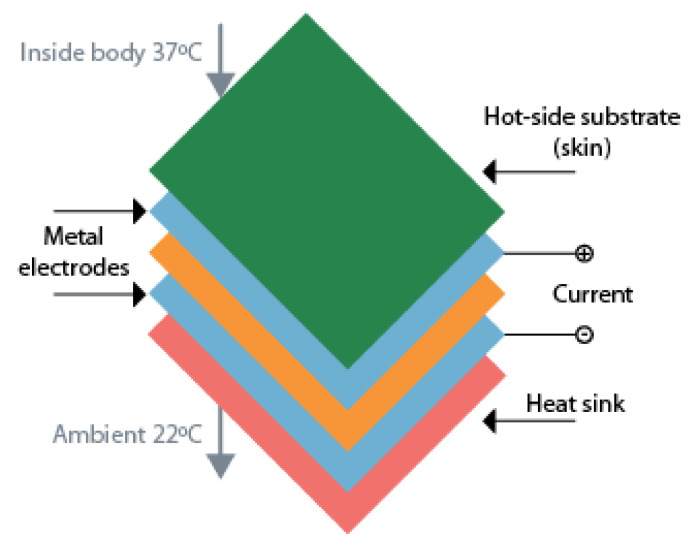
Schematic of a thermoelectric energy harvester applied to human body. Adapted from [74].

**Figure 10 sensors-20-06420-f010:**
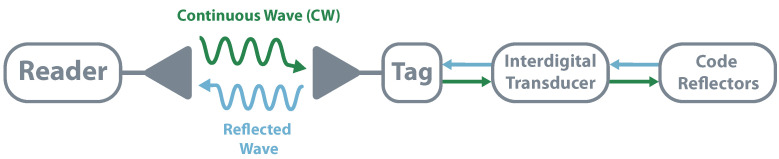
Schematic of a Time Domain Reflectometry system using Surface Acoustic Wave.

**Figure 11 sensors-20-06420-f011:**
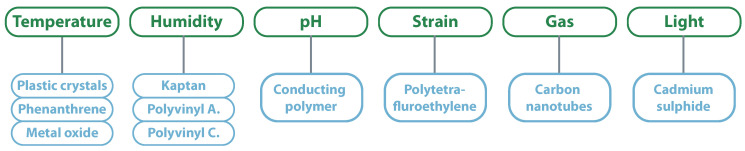
Classification of smart materials.

**Figure 12 sensors-20-06420-f012:**
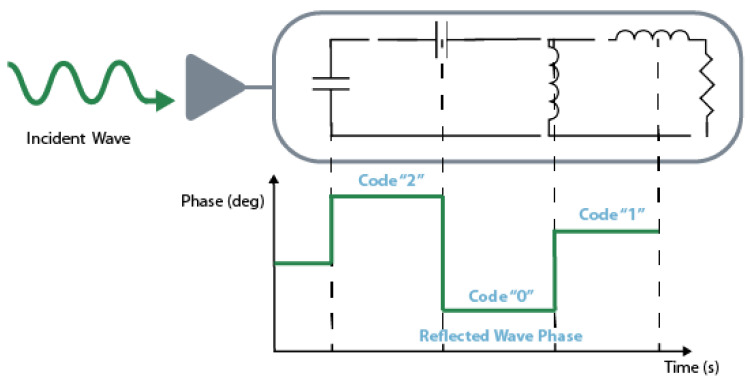
Encoding based on the phase modulation. Adapted from [105].

**Figure 13 sensors-20-06420-f013:**
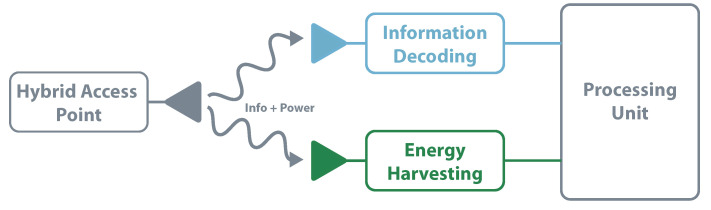
SWIPT—independent receiver architecture.

**Figure 14 sensors-20-06420-f014:**
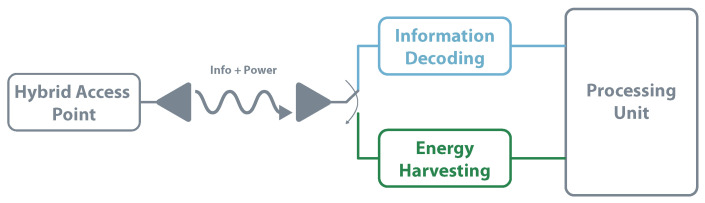
SWIPT—time switching architecture.

**Figure 15 sensors-20-06420-f015:**
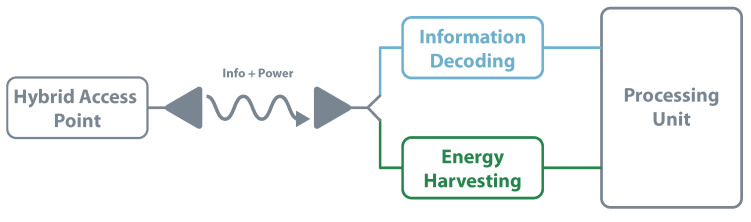
SWIPT—power splitting architecture.

**Figure 16 sensors-20-06420-f016:**
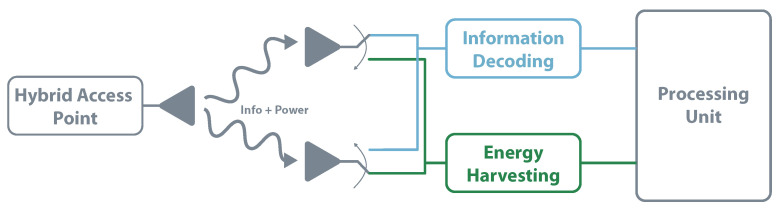
SWIPT—antenna switching architecture.

**Figure 17 sensors-20-06420-f017:**
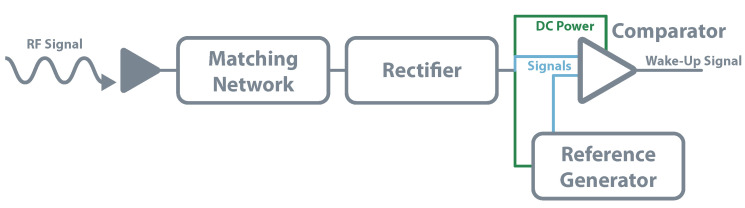
WUR—passive architecture.

**Figure 18 sensors-20-06420-f018:**
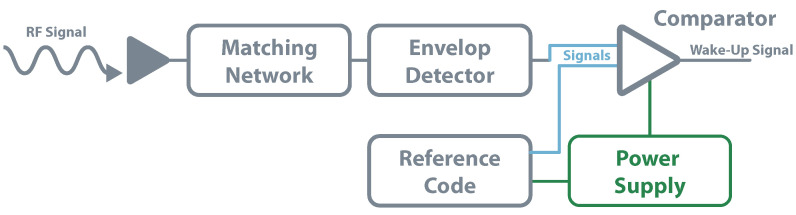
WUR—active architecture.

**Figure 19 sensors-20-06420-f019:**
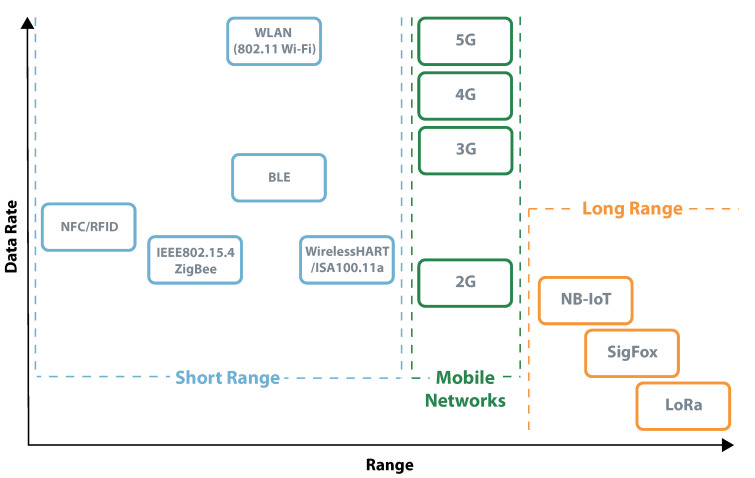
Data rate vs. Range—active radio communication technologies.

**Figure 20 sensors-20-06420-f020:**
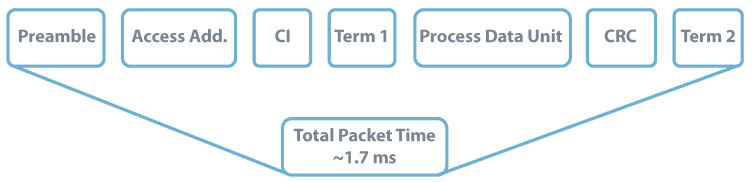
BLE Packet Structure-Payload = 16 bytes. Data from source [138].

**Figure 21 sensors-20-06420-f021:**
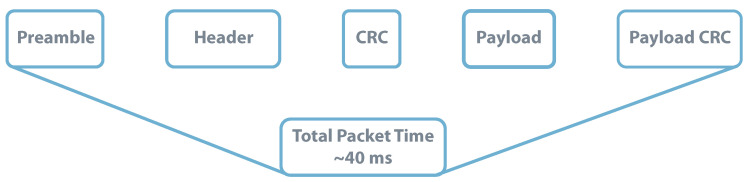
LoRa Packet Structure-SF = 7, BW = 125 kHz, CRC = 0 and Payload = 16 bytes. Data from source [139].

**Table 1 sensors-20-06420-t001:** Comparison between communication interfaces

	Frequency Band	Range	Data Rate	Consumption
NFC	13.56 MHz	10 cm	400 kbit/s	60 mA
BLE	2.4 GHz	1 Km	2 Mbit/s	7.5 mA
ZigBee	2.4 GHz	100 m	250 kbit/s	10 mA
Wi-Fi	2.4, 3.6, 5 GHz	100 m	1.3 Gbit/s	240 mA
Mobile Networks (2G/3G)	900 MHz, 1.8 GHz, 2.1 GHz	-	100 kbit/s (2G), 8 Mbit/s (3G)	330 mA
Sigfox	868 MHz	10 Km	100/600 bit/s	49 mA
LoRa	868 MHz	10–15 Km	50 kbit/s	39 mA
NB-IoT	700, 800, 900 MHz	-	200/50 kbit/s	220 mA

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
