# Peer review of "Challenges in Resource-Constrained IoT Devices: Energy and Communication as Critical Success Factors for Future IoT Deployment"

_sensors, 2020, doi:10.3390/s20226420_

Round 1
Reviewer 1 Report
Dear authors since this is a review work on IoT And the topic is certainly of interest for the researchers community involved in IoT development.
the following suggestions aimed at improve the quality of the manuscript should be taken into account by you.
1) Passive tag, in this section the modulated scattering technique MST is completely omitted, it is quite similar to RFID since it is based on the use of an antenna closed by means of a suitable switch on different resistive load. Please take a look at the pioneering work of proff. Bolomey and of the other researchers involved in the development of MST systems. MST can provide very long range rfid
2) the section devoted to chipless tag need to be improved it turn out to be too much limited, please insert further consideration, comments, comparisons between chipless rfid and IoT system in time as well as frequency domain.
3) also the wpt systems description should be improved in my opinion it turn out to be too much limited and it should be extended. In particular providing more emphasis on microwave wpt systems for long range to provide power supply to IoT systems.
4) this is a review paper so the reference section should be extended in this form it is too much limited for a review paper.
Reviewer 2 Report
Challenges in resource-constrained IoT devices: energy and communication as critical success factors for future IoT deployment
SUMMARY
The article provides a comprehensive review of resource-constrained IoT devices grouping them into 3 big categories: passive, semi-active, and active. The paper mostly includes what appears to be a possibly useful, detailed and comprehensive review of the future technological challenges that should overcome IoT devices in terms of energy and communication.
I think that the manuscript is well structured and written in a style comprehensible to the average reader, providing a good starting point and an initial set of references for readers and scientific community to delve further. Additional contents (whether original or adapted) such as Tables and Figures are also worth of being included, as they help to understand, clarify and/or compile and summarize information already provided in text.
Overall, I recommend publishing this article in MDPI Sensors Journal. However, in the following lines I formulate some comments/questions that should be solved to even improve the quality of the work.
Keywords
- Intenet of Things (IoT) --> Internet of Things
Section 1 – Introduction
- “to the internet.” --> “to the Internet”
- “energy, communication and, compatibility and standardization” -- > “energy, communication, and compatibility and standardization”
- “Currently, the used architectures for data centers are not prepared to deal with the quantity and diversity of data that billions of heterogeneous devices can generate. Also, there is a need for new intelligent analytics mechanisms” --> I would include some reference to new IoT architectures, such as Fog Computing, with intermediate devices conducting processing tasks. Machine Learning applied to IoT would be also worth to be mentioned here.
- “Regarding communication systems, there are multiple alternatives, such as Wi-Fi, Bluetooth, and Low PowerWide Area Network (LPWAN) which use active radios.” --> I would also add here Wireless Sensor Networks (WSN) and cellular systems to show the full spectrum of wireless communications for IoT
- “and Section 4 resumes some alternatives for active systems. The document ends with its conclusions in Section 6.” --> Nothing to say about Section 5?
Section 2 – Passive Systems
- “The way to do not use” --> “The way to not use”
- “choosing the best energy source always depend on the final application” --> “choosing the best energy source always DEPENDS on the final application”
- “Nevertheless, the scarcity of works, it was possible to find some good examples.” --> “DESPITE / IN SPITE OF the scarcity of works, it was possible to find some good examples.”
- “The target cost is less than 0,01,” --> Less than 0.01 what? Euro? Dollar? (Note also the use of the point instead of the comma as decimal separator)
- “The main application for SAW devices was been identification proposes” --> “The main application for SAW devices HAS/HAD been identification proposes”
Section 4 – Active Systems
- “The following short-range interface under analyzes is ZigBee,” --> I understand the popularity and well-knowledge of ZigBee, but in this context, where only physical parameters are taken into consideration, we should better talk about IEEE 802.15.4 instead. In fact, IEEE 802.15.4 is the standard which defines the physical (PHY) and medium access control (MAC) layers of some different specifications such as ZigBee, Thread or 6LoWPAN.
- “In its 802.11ac version, it can theoretically reach 1.3Gbit s?1, having being named Gigabit Wi-Fi. It can operate at 2.4,3.6 and 5GHz” --> I would not include here the 3.6 GHz band of WiFi, as it is only used by IEEE 802.11y. If so, other available (or under study) frequency bands should also be included, such as sub-1GHz, 6 GHz and 60 GHz.
- ”which allow LoRa chips to have a sensitivity of -146 d” --> ”which allow LoRa chips to have a sensitivity of -146 dBm”
- “NB IoT is a technology” --> “NB-IoT is a technology”
- “For simplification, it was considered that each data transmission takes 1 second regardless of the device in use, which prejudices the BLE-based solutions once it has higher data rates than the LoRa-based ones.” --> I think that the results associated to the real application scenario using BLE and LoRa should be improved. For instance, the 1-second simplification is too drastic. I would define a packet length, then compute the packet transmission time for each technology taking into consideration the data rate, and lastly compute the energy consumption.
- On this matter, be careful with the notation in equations (1) and (2) and complementary text; you are not computing “Energy”, expressed in Joules (J) or Watt-hour (W·h), but “Electric charge”, expressed in Coulombs ( C ) or Ampere-second (A·s).
- I would also include all steps to fulfill equations (1) and (2); i.e., by saying what represents each number and including its corresponding unit (currently, different units are mixed in the same equation).
Reviewer 3 Report
- Actually is not the "bandwidth of IoT terminals could vary from a few kbps to several Mbps". It is the "data rate" or "channel throughput". The bandwidth is measured in [Hz]. Please review.
- In this case "achieve efficient use of power and bandwidth", if bandwidth is related to "data rate", please replace. If it is related to "efficient use of bandwidth" than it is spectral efficiency [kbits/MHz/km2], also an important measure of IoT systems. Actullay, spectral efficiency expressed in terms of [kbits/MHz/km2] joins non-correlated measures as data rate, channel bandwidth and coverage (range or distance as it was indicated). Please review.
- There are not only "commercial protocols" like Bluetooth or LoRa (see line 61). There are also industial IoT protocols like ISA100 and WirelessHART. Actually the major security, communications and energy issues are related to industrial IoT (IIoT) devices, not to the commercial ones.
- Please review and extend the paper considering industial IoT devices and networks, and their relations to the five or two selected metrics considered to be relevant in this paper.
- The reviewer considers relevant in the context of this paper the following publication:
- E. Jecan, C. Pop, Z. Padrah, O. Ratiu and E. Puschita, "A dual-standard solution for industrial Wireless Sensor Network deployment: Experimental testbed and performance evaluation," 2018 14th IEEE International Workshop on Factory Communication Systems (WFCS), Imperia, 2018, pp. 1-9, doi: 10.1109/WFCS.2018.8402360.
- Please consider and review the above references (1 to 3).
- From a communication protocols perspectives, the details presented in the paper are only related to the physical layer (basic modulation schemes, and transmission concepts and block diagrams). The reviewer considers all these aspects out of scope of an IoT devices review article focused on energy and communications. Form the reviewers perspective, communication protocols review is mandatory to be presented in a IoT devices review article.
- The data rate vs. range (see Fig. 19) is not objective. For example, the maximum data rate in 2G networks is 14.4 kbps. As this graph presents, the 2G data rate is comparable with the WLAN (802.11) network. Actually is not WiFi - which is the Wireless Fidelity Forum, it is WLAN. The range is highly dependent to the receiver sensitivity (cell size) and data rate. Please review.
- Figure 20 actually is a table. Please review!.
- Figure 20 must include ISA100 and WirelessHART industrial communication protocols. The reviewer considers that "interfaces" must be replaced by "protocols" as all this technologies are standardized as protocols not as interfaces. Related to Wi-Fi please consider note 8. WiFi is not a protocol, is a forum!
- Moreover, "cellular" is related to a network planning aspect not to a technology. It cannot be presented alongside ZigBee and LoRa ... Maybe GMS, UMTS, LTE must be considered as standards. Please review!
- Please point out the originality of this work.
- Please indicate how can benefit from the results of this work.
Round 2
Reviewer 1 Report
The revised version of the manuscript has been improved I have no further concerns
Reviewer 3 Report
All comment were addressed.